# Relationship between Duffy Genotype/Phenotype and Prevalence of *Plasmodium vivax* Infection: A Systematic Review

**DOI:** 10.3390/tropicalmed8100463

**Published:** 2023-09-30

**Authors:** Yelson Alejandro Picón-Jaimes, Ivan David Lozada-Martinez, Javier Esteban Orozco-Chinome, Jessica Molina-Franky, Domenica Acevedo-Lopez, Nicole Acevedo-Lopez, Maria Paz Bolaño-Romero, Fabriccio J. Visconti-Lopez, D. Katterine Bonilla-Aldana, Alfonso J. Rodriguez-Morales

**Affiliations:** 1Fac Ciències Salut Blanquerna, University Ramon Llul, 08022 Barcelona, Spain; ypicon@unab.edu.co; 2Epidemiology Program, Department of Graduate Studies in Health Sciences, Universidad Autónoma de Bucaramanga, Bucaramanga 44005, Colombia; ilozada@unab.edu.co; 3Medical and Surgical Research Center, Future Surgeons Chapter, Colombian Surgery Association, Bogotá 10002, Colombia; efeso17@yahoo.es (J.E.O.-C.); nicol_1008@hotmail.com (N.A.-L.); mbolanor1@unicartagena.edu.co (M.P.B.-R.); 4Department of Inmunology and Theranostics, Arthur Riggs Diabetes and Metabolism Research Institute, Beckman Research Institute of the City of Hope, Duarte, CA 91007, USA; jessicamolinafranky@gmail.com; 5Molecular Biology and Inmunology Department, Fundación Instituto de Inmunología de Colombia (FIDIC), Bogotá 10001, Colombia; 6School of Medicine, Fundación Universitaria Autónoma de las Américas-Institución Universitaria Visión de las Américas, Pereira 660003, Colombia; domenica.acevedo@uam.edu.co; 7Sociedad Científica de Estudiantes de Medicina—UPC, Lima 13008, Peru; fabricciovisco@gmail.com; 8Research Unit, Universidad Continental, Huancayo 12000, Peru; 9Clinical Epidemiology and Biostatistics Master Program, Universidad Cientifica del Sur, Lima 15067, Peru; arodriguezmo@cientifica.edu.pe; 10Gilbert and Rose-Marie Chagoury School of Medicine, Lebanese American University, Beirut P.O. Box 36, Lebanon

**Keywords:** *Plasmodium vivax*, *vivax* malaria, Duffy blood group system, *Plasmodium* Duffy antigen binding protein, prevalence

## Abstract

The Duffy protein, a transmembrane molecule, functions as a receptor for various chemokines and facilitates attachment between the reticulocyte and the *Plasmodium* Duffy antigen-binding protein. Duffy expression correlates with the Duffy receptor gene for the chemokine, located on chromosome 1, and exhibits geographical variability worldwide. Traditionally, researchers have described the Duffy negative genotype as a protective factor against *Plasmodium vivax* infection. However, recent studies suggest that this microorganism’s evolution could potentially diminish this protective effect. Nevertheless, there is currently insufficient global data to demonstrate this phenomenon. This study aimed to evaluate the relationship between the Duffy genotype/phenotype and the prevalence of *P. vivax* infection. The protocol for the systematic review was registered in PROSPERO as CRD42022353427 and involved reviewing published studies from 2012 to 2022. The Medline/PubMed, Web of Science, Scopus, and SciELO databases were consulted. Assessments of study quality were conducted using the STROBE and GRADE tools. A total of 34 studies were included, with Africa accounting for the majority of recorded studies. The results varied significantly regarding the relationship between the Duffy genotype/phenotype and *P. vivax* invasion. Some studies predominantly featured the negative Duffy genotype yet reported no malaria cases. Other studies identified minor percentages of infections. Conversely, certain studies observed a higher prevalence (99%) of Duffy-negative individuals infected with *P. vivax.* In conclusion, this systematic review found that the homozygous Duffy genotype positive for the A allele (FY*A/*A) is associated with a higher incidence of *P. vivax* infection. Furthermore, the negative Duffy genotype does not confer protection against vivax malaria.

## 1. Introduction

Malaria, one of the most prevalent and potentially deadly infectious diseases worldwide, is estimated to have caused 229 million cases and 409,000 deaths in 2019, according to the World Health Organization (WHO) [1]. The WHO World Malaria Report 2022 suggested that there were 247 million global cases and 619,000 deaths in 2021 [2]. In the United States of America, around 2000 new cases are reported annually, predominantly due to imported malaria from countries with high transmission rates, such as sub-Saharan Africa and Southeast Asia [3].

The illness occurs when parasites belonging to Plasmodium species invade and multiply in red blood cells, causing changes that could be life-threatening. The infection occurs when infected female *Anopheles* mosquitoes bite humans. It is recognised that five species, namely *P. falciparum*, *P. vivax*, *P. ovale wallikeri*, *P. ovale curtisi*, *P. malariae*, and *P. knowlesi*, can lead to health issues in humans. Additionally, infected individuals can remain infected for years and act as asymptomatic carriers for years without causing any apparent illness. The most significant and widespread species are *P. falciparum* and *P. vivax*, with the former causing more severe disease and the latter having a global distribution [4].

The Duffy glycoprotein, a non-selective receptor, interacts with various chemokines such as interleukin-8 (IL-8) and melanoma growth-stimulating activity (MGSA) in the CXC group, as well as monocyte chemotactic protein-1 (MCP-1) and CCL5 in the CC group [5]. These interactions attract monocytes, memory CD4+ T lymphocytes, and eosinophils. The Duffy protein, encoded by the ACKR1 gene (also known as Duffy antigen receptor for chemokines [DARC]), is located on chromosome 1 at 1q23.2 [6,7]. The interaction with DARC is essential for the invasion of *Plasmodium vivax.* Consequently, the absence of this protein is considered a preventive factor against the invasion of these merozoites into reticulocytes. Most African populations in sub-Saharan Africa either exhibit low expressions of the Duffy antigen or are Duffy-negative, which correlates with a lower proportion of *P. vivax* malaria in this region. Furthermore, the heterozygous phenotypes Fy(a+b−) and Fy(a−b+) (Duffy-positive heterozygotes) may exhibit a certain degree of resistance to parasite infection compared to the Fy(a+b+) phenotype [7]. However, insufficient global data currently exist to support this phenomenon.

To achieve the WHO’s goal of controlling and eradicating *P. vivax* malaria [1], it is imperative to design and conduct studies identifying the relationship between the Duffy genotype/phenotype and the prevalence of *P. vivax* infection [8], given that more than one-third of the global population is exposed to it [9]. Recent publications have reported cases of *P. vivax* malaria in Duffy-negative patients [8], contradicting the long-established belief that the Duffy antigen is essential for the invasion of *P. vivax* parasites into reticulocytes [10,11,12]. This finding holds significant implications for understanding the disease and designing new drugs. Thus, this study aims to evaluate the relationship between the Duffy genotype/phenotype and the prevalence of *P. vivax* infection.

## 2. Methods

### 2.1. Registration of the Systematic Review Protocol

The protocol of systematic review was registered at PROSPERO: CRD42022353427.

### 2.2. Guideline of Reporting Systematic Review

The reports of this systematic review followed the PRISMA statement [13].

### 2.3. Research Question

Is the Duffy-negative genotype/phenotype a protective factor in the population susceptible to *P. vivax* infection compared to the Duffy-positive population?

### 2.4. Search Strategy

The search was conducted in Medline, Web of Science, Scopus, and Scielo. The selected terms were Duffy blood group system; ACKR1 protein, human; Duffy antigen binding protein, *Plasmodium; Plasmodium*; and Malaria. These terms were combined using the boolean operators “OR” and “AND”. Due to the broadness of the search results, additional filters were applied to select only those manuscripts corresponding to clinical studies, clinical trials, controlled clinical trials, multicenter studies, observational studies, and randomised controlled trials. The searches were slightly modified according to each database.

### 2.5. Eligibility Criteria and Study Selection

The inclusion criteria for the studies were as follows: (1) consideration was given to descriptive observational manuscripts, analytical observational studies, experimental studies, and quasi-experimental studies; (2) texts in Spanish, English, and Portuguese that matched the search equation were included; (3) texts referring to the Duffy genotype, Duffy binding protein, or DARC and their relationship with *P. vivax* or other *Plasmodium* infection, within the specified time frame, were included; (4) articles with a methodological quality score higher than 60% according to the STROBE (Strengthening the Reporting of Observational Studies in Epidemiology) assessment tool were included; (5) articles published within the last ten years were included; and (6) full-text articles were considered.

Conversely, articles meeting the following criteria were excluded: (1) articles that did not provide data on events related to the Duffy genotype/phenotype and malaria or addressed infection or infection by a microorganism other than *Plasmodium*; (2) posters, abstracts, topic reviews, and informal texts were not included in the review; and (3) integrative reviews and systematic/meta-analyses were also excluded.

### 2.6. Quality of the Included Studies

The articles’ quality was assessed using the STROBE tool [14], considering that the manuscripts were observational studies. The STROBE tool comprises a checklist of 22 items for reporting this type of manuscript. A minimum compliance threshold was established, requiring the fulfilment of at least 60% of the checklist items, equivalent to meeting at least 14 out of the 22 points. Furthermore, the study’s risk of bias, certainty, and importance was evaluated using the GRADE tool [15].

## 3. Results

### 3.1. Search Results

At the end of the database search, along with reverse searching based on article references, 3600 manuscripts were obtained. The distribution was as follows: Medline/PubMed contained 3101 texts, Web of Science had 263 texts, Scopus contained 230 texts, Scielo contained 1 text, and 5 texts were obtained through reverse searching. The review process involved discarding duplicates, screening titles and abstracts, and concluding with full-text reading. Ultimately, this systematic review included 34 articles [7,10,11,12,16,17,18,19,20,21,22,23,24,25,26,27,28,29,30,31,32,33,34,35,36,37,38,39,40,41,42,43,44,45] (Figure 1).

### 3.2. Quality of the Included Studies

According to the STROBE criteria that assessed the study’s quality with an overall score of 22 items, 19 studies achieved a score higher than 85% (*n* = 19 items). In comparison, three studies obtained a lower score, with a compliance rate of 72.7% (*n* = 16 items) of the items (Figure 2). During the application of the GRADE tool, a risk of serious bias was found in 70.5% (*n* = 24) of the studies, and the same percentage of studies exhibited moderate certainty of the evidence (70.5%; *n* = 24) (Table 1).

### 3.3. General Characteristics and Relationship between Duffy Genotype/Phenotype and Prevalence of Plasmodium vivax

According to the distribution by continent, 21 studies were conducted in Africa. In Ghana, a study analysed 952 subjects (845 symptomatic and 107 healthy individuals). Among these cases, 542 tested PCR-positive for *P. falciparum*, 1 for *P. malariae*, and 2 for *P. ovale.* The FY*B^ES^/B^ES^ genotype corresponding to the Duffy-negative phenotype Fy(a−b−) was found in 862 subjects, while 53 individuals had the FYA/B^ES^ genotype with the Fy(a+b−) phenotype. Additionally, 22 individuals had the FYB/B^ES^ genotype with the Fy(a−b+) phenotype and 15 had the FYA/B genotype with the Fy(a+b+) phenotype. No cases of *P. vivax* malaria were confirmed through PCR, and the limited evidence of *P. vivax* pathology was attributed to the high frequency of the FYES allele [10].

Sudan was the site of five studies. The first study analysed 412 blood samples from patients infected with *P. vivax*, including 155 subjects from Khartoum, the Nile River, and New Halfa. This study also included samples from Ethiopia, specifically from the Jimma, Gojeb, and Arjo regions, where 150 subjects with *P. vivax* malaria were studied. Of these samples, 305 were identified as Duffy-positive and 107 as Duffy-negative; from Ethiopia, 150 were Duffy-positive and 83 were Duffy-negative, while from Sudan, 155 were Duffy-positive and 24 were Duffy-negative samples. Among individuals with Duffy-negative status, the infection rate was highest at 17.2%, while among Duffy-positive individuals, the highest rate was 30.7% [11]. The second study, conducted in Khartoum, New Halfa, and the Nile River, recruited patients with symptomatic malaria from primary care centres. A total of 992 microscopy-positive samples were taken, confirming 186 cases as mono-infection by *P. vivax* and identifying 4 cases as mixed infections by *P. vivax*/*P. falciparum*, as determined through PCR. Of the 190 samples with *P. vivax* malaria and 67 healthy individuals, 129 cases (67.9%) were Fy(a−b+), 14.2% were Fy(a+b−), and 17.9% were Fy(a−b−). The Fy(a+b+) phenotype was not detected. Among the healthy individuals, 45 (67.1%) were Fy(a−b−), 29.9% were Fy(a−b+), and only 3% had the Fy(a+b−) phenotype. The Fy(a−b+) and Fy(a+b−) phenotypes were significantly higher in *P. vivax*-infected patients than in healthy individuals (*p* < 0.01). Conversely, Duffy-negative individuals (Fy(a−b−)) exhibited a significantly lower proportion of *P. vivax* infection (*p* < 0.01). The most prevalent phenotype was Fy(a−b+). In New Halfa, 62.5% (*n* = 25/40) of *P. vivax* samples were classified as Fy(a−b+). The prevalence of infection in Khartoum and the Nile River showed a similar trend, with Fy(a−b+) accounting for 80.2% and 46.9%, respectively. Fy(a−b−) individuals had significantly lower levels of *P. vivax* parasites compared to Fy(a+b−) and Fy(a−b+) individuals (*p* < 0.001) [12]. The third study collected 63 blood samples, of which 42 demonstrated *P. vivax* infection. These samples were obtained from patients in different areas of Sudan from 2014 to 2016. Thirty-five (83.3%) were identified as Duffy-positive (ten homozygotes and twenty-fiveheterozygotes), while seven (16.7%) were Duffy-negative. This study detected seven cases of *P. vivax* in Duffy-negative individuals, characterised by mutations in six PvRBP haplotypes. However, it was observed that five PvRBP haplotypes were shared between Duffy-negative and Duffy-positive individuals, except for one haplotype exclusive to Duffy-negative individuals [42]. Lo et al. [35], who investigated this phenomenon in Sudan, Botswana, and Ethiopia, demonstrated that in Botswana, 83.5% (*n* = 147/176) of febrile patients were Duffy-negative. Among the febrile patients in Kweneng East, 3% (*n* = 9/301) tested positive for *P. vivax*, comprising eight Duffy-negative homozygotes (CC) and one Duffy-positive heterozygote (TC). In Tutume, 6.8% (*n* = 12/176) of febrile patients were identified with *P. vivax*, with 10 of them being Duffy-negative. Conversely, the proportion of febrile individuals from Ethiopia exhibited a Duffy-negative pattern of 35.9% (*n* = 235/655). In Bonga, 30.3% (*n* = 125/413) of febrile patients were diagnosed with *P. vivax*, and 3.2% (*n* = 4/125) were Duffy-negative. In Sudan, 831 samples were collected, confirming 101 cases of *P. vivax* infection, with 7 occurrences in Duffy-negative individuals [35].

The fourth study in this country was conducted in Gezira, central Sudan, where 126 patients with suspected malaria were analysed from October to December 2009. *P. vivax* infection was identified in 48 subjects, representing 38% of the samples, using PCR. Gezira state is characterised by a seasonal and unstable transmission of *P. vivax* malaria, with the rainy season starting in July and ending in October and an annual rainfall ranging between 140 and 225 mm. Regarding gender, 54.2% were women, and 79.2% were under ten years old [18].

In Ethiopia, a study involving 178 individuals (145 with symptomatic *P. vivax* malaria and 33 asymptomatic) revealed that 101 (69.7%) of the symptomatic individuals had heterozygosity with a silenced Duffy allele (FY*A/B^ES^ or FYB/B^ES^), 2 (1.4%) were homozygotes for Duffy negativity (FYB^ES^/B^ES^), 17 (11.7%) were homozygotes for Duffy positivity (FyA/A or FyB/B), and 25 (17.2%) were heterozygotes positive for Duffy (FyA/*B) [26]. Another study, which assessed 416 symptomatic malaria patients, identified that 94 (23%) samples exhibited homozygosity for the CC genotype at nucleotide position -33 (indicating Duffy negativity). The Duffy-positive samples numbered 322 (77%), with 108 (26%) being TT homozygotes and 214 (51%) being CT heterozygotes. Two of the ninety-four Duffy-negative samples tested positive for *P. vivax*, indicating *P. vivax*/*P. falciparum* mixed infections. Among the samples from healthy individuals, 35.6% (*n* = 139/390) were found to be Duffy-negative, a proportion significantly higher compared to the case group (*p* < 0.0001). Overall, the prevalence of *P. falciparum* exceeded that of *P*. *vivax* in both Duffy-positive and Duffy-negative subjects [23]. In the same country, in 2009, Woldearegai et al. [21], in an analysis of 205 *Plasmodium*-positive samples, revealed the presence of the FYA/A genotype in 1 individual (0.9%) in Jimma, FYA/B in 11 subjects (11.2%) in Harar, and 14 (13.1%) in Jimma. The FYA/B^ES^ genotype was detected in 4 individuals (4.1%) in Harar and 18 (16.8%) in Jimma. Conversely, the most prevalent genotype was FYB/B, identified in 51 cases (52%) in Harar and 21 (19.6%) in Jimma. The FYB/*B^ES^ genotype was observed in 15 subjects (15.3%) in Harar and 29 (27.1%) in Jimma. Regarding the phenotype, the Duffy-positive phenotype was identified in 82.7% and 77.6% of individuals in Harar and Jimma, respectively, while the Duffy-negative phenotype prevalence in these locations was 17.3% and 22.4%. In Senegal, a study was conducted on 48 children who were classified as Duffy-negative (FYB^ES^/B^ES^), including five with *P. vivax* infection, which confirmed a surprisingly high proportion (20.3%) of *P. vivax* malaria among children with a negative phenotype [41].

In the Democratic Republic of the Congo, 292 dried blood samples from children who participated in the Demographic and Health Survey of 2013–2014 were analysed. It was confirmed that 14 *P. vivax*-positive children exhibited a negative Duffy phenotype. Fourteen cases of *P. vivax* malaria were identified through PCR, and nine were co-infected with *P. falciparum.* All *P. vivax* cases occurred in rural households, and only five out of the fourteen cases were reported using long-lasting insecticidal nets. Regarding socio-economic status, nine cases were classified within the poorest population [37]. Another study in the Congo assessed men and women aged 15 to 59 years and 15 to 49 years, respectively. Out of the 17,972 samples screened for *P. vivax* infection, 579 tested positive through PCR, and 534 were confirmed using nested PCR (92.2%), indicating strong agreement (Kappa = 0.80, *p* < 0.05). Almost all individuals affected by *vivax* malaria had a Duffy-negative status (*n* = 464/467; 99.36%). However, no further demographic data were available for this study [40].

In Nigeria, a study was conducted on blood samples from 242 subjects aged 25 years, of whom 55% were women. Malaria caused by *P. falciparum* was found in 133 individuals, while 6 cases showed a mixed infection of *P. falciparum*/*P. ovale*, 3 cases were attributed to *P. vivax*, 1 case had co-infection of *P. falciparum*/*P. vivax*, and 1 case involved *P. malariae* and *P. ovale*. The analysis revealed that a single cytosine at nucleotide position 33 was present in four patients who had experienced *P. vivax* malaria. This finding confirmed the absence of Duffy gene expression in their cells, indicating a Duffy-negative genotype [45]. Another study specifically targeted the states of Lagos (hypoendemic with a prevalence of 1.9%) and Edo (mesoendemic with a prevalence of 35%). It included patients over two years old presenting clinical symptoms of malaria. A total of 2376 patients were enrolled, and malaria was confirmed in 436 samples. The mean age was 23 years, and 55% were women. *Plasmodium* RNA was amplifiable in 58.7% (*n* = 256/436) of the subjects, with 110 from Edo and 146 from Lagos. The majority of cases were attributed to *P. falciparum* as a mono-infection (85.5%; *n* = 219/256; 97 from Edo and 122 from Lagos) or mixed with *P. malariae* (6.3%; *n* = 16/256), *P. vivax* (1.6%; *n* = 4/256), or *P. ovale* (1.2%; *n* = 3/256) [34].

In Cameroon, an analysis was conducted on febrile outpatient patients of all ages who sought consultation at the Santchou, Dschang, and Kyéossi health centres. The individuals included were 400, 500, and 101, respectively. A total of 287 cases of *P. falciparum* infection were detected using PCR, along with 142 cases of *P. vivax*, 2 cases of *P. ovale*, and 3 cases of *P. malariae*. Additionally, there were 37 cases of co-infection with *P. falciparum*/*P. vivax*, 2 cases of *P. falciparum*/*P. ovale*, 4 cases of *P. falciparum*/*P. malariae*, and 2 cases of *P. vivax*/*P. malariae* [43]. In Dschang, a total of 484 samples were obtained from febrile outpatient patients by other researchers. Malaria parasite DNA was identified in 70 samples (14.5%), including 68 cases of mono-infection by *Plasmodium* (42 cases of *P. falciparum*, 25 cases of *P. vivax*, and 1 case of *P. malariae*), as well as 2 cases of co-infection with *P. falciparum*/*P. vivax*. Among the affected individuals, 57.1% were male, and the median age was 24. Specifically, 74.3% originated from an urban population area. In this case, a 2.3-times higher likelihood of testing positive for *Plasmodium* via PCR (95% CI: 1.39–3.89; *p* = 0.0014) was found to be associated with being male. Two homozygous Duffy-positive genotypes (−33 TT), two heterozygotes (−33 TC), and 224 Duffy-negative individuals (−33 CC) were identified by other researchers. All individuals with *P. vivax* demonstrated a Duffy-negative genotype. The overall frequency of the −33T allele was 1.3%, corresponding to a frequency of 1.7% (*n* = 4/228) of positive Duffy phenotypes (homozygotes and heterozygotes) [33]. A third study involved 485 symptomatic patients who attended hospitals in five different areas in the country’s southern region revealed that among 201 malaria cases, including 8 cases of *P. vivax*, all 8 patients exhibited the −33 CC mutation, indicating a Duffy-negative status in all 8 native Cameroonians [20]. In another study conducted explicitly in Bolifamba, a rural multiethnic environment situated at an altitude of 530 m on the eastern slope of Mount Cameroon, samples were collected from 269 individuals. The results revealed a *Plasmodium* prevalence of 32.3%. Exclusive or concomitant *P. vivax* infections accounted for 14.9% (*n* = 13/87) of the cases, as established through PCR and microscopic examination. Moreover, 50% of individuals (*n* = 6/12) affected by *P. vivax* malaria were also negative for the Duffy receptor [19].

In Mali, blood samples were collected from 300 children aged 0 to 6 years, revealing 25 cases of malaria caused by *P. vivax* and 109 cases caused by *P. falciparum*. Among the 25 cases, a Duffy-negative genotype with the presence of the T to C mutation in the GATA1 5′ binding site of the open reading frame was identified in all cases. However, no information was found regarding the population characteristics [38].

In Namibia, a study involved 952 individuals under nine, of whom 52.6% were females and 63.4% were afebrile. Most cases involved mono infections by *P. falciparum* (*n* = 23), and *P. vivax* infected 3individuals. Additionally, there were four co-infections by *P. falciparum*/*P. vivax* and three by *P. falciparum*/*P. ovale*. Five participants with *P. vivax* tested negative for the Duffy gene mutation. Among 9 out of 41 participants not infected with *Plasmodium* and 7 out of 28 participants with *P. falciparum*, the FYA genotype (Duffy-positive) was present. There was a C136 G > A mutation in exon two that was present in all patients with the *P. vivax* infection (*n* = 5/7) [31].

In Madagascar, a study focused on 129 individuals seeking antimalarial treatment between 2015 and 2017, all of whom had malaria caused by *P. vivax*. Fifty-five exhibited a Duffy-positive genotype, with 56% being heterozygotes and 44% being homozygotes for DARC gene expression [39]. Another study conducted in the western part of the country, specifically in Tsiroanomandidy, in the Bongolava region, analysed 2143 subjects (53% females, average age of 19.6 years). This rural area is endemic to malaria caused by *P. falciparum* and *P. vivax*. Symptomatic malaria cases were sporadic, with only 11 individuals affected. *Plasmodium* invasions were generally submicroscopic, and 82.8% went undetected by microscopy (2.4% prevalence with microscopy-positive results [*n* = 49] vs. 13.8% prevalence with PCR-positive results [*n* = 285]). It was reported that out of 1878 individuals, the most frequent allele was the silent erythrocyte allele FYB^ES^, followed by FYA and FYB. Approximately 48.7% of the subjects had a Duffy-negative phenotype. Among Duffy-positive individuals (51.3% of the total population), Fy(a+b−) was the most common phenotype at 34.5%, followed by Fy(a−b+) at 11.6%, and Fy(a+b+) at 5.2%. The number of Duffy-positive individuals with *P. vivax* was 86 (8.9%), while there were 44 (4.8%) Duffy-negative individuals. Thus, it was determined that the risk of malaria for Duffy-negative hosts was half that of Duffy-positive hosts (prevalence of 4.8% vs. 8.9%; OR 0.52; 95% CI: 0.35–0.75; *p* < 0.001). There were no statistically significant differences in the likelihood of infection between homozygous Duffy-positive and heterozygous Duffy individuals (*p* = 0.429), although heterozygotes had a slightly lower infection prevalence (8.5% vs. 10.2%). Finally, no association was found between Duffy blood type and *P. falciparum* malaria [36].

In Cambodia and Madagascar, among 453 individuals with *P. vivax* malaria, the genotype was determined for 119 individuals who exhibited Duffy positivity with the T-33C substitution T/T, indicating their homozygous Duffy-positive status [39]. In a separate study conducted by Popovici et al. in the same country, it was reported that all genotyped reticulocytes demonstrated Duffy positivity, with most of them (*n* = 16/22) being homozygous FY*A/A. At the same time, the remaining (*n* = 6/22) were heterozygous FYA/*B [30].

In India, on the Asian continent, 909 outpatient malaria patients and 2478 healthy individuals were recruited between June and December 2015. The median age in the cases was 26 years, while in the controls, it was 30 years. Males constituted 92.8% of the cases and 57.5% of the controls. Only 4.2% of the cases reported using mosquito nets, compared to 42.4% in the control group. Among the patients, malaria caused by *P. vivax* (70%, *n* = 633) was more prevalent than malaria caused by *P. falciparum* (9%, *n* = 82) and the combination of *P. vivax*/*P. falciparum* (21%, *n* = 194). All its cases and controls revealed the exclusive occurrence of DARC 298A when 125A was also present, specifically in FYB (*p* < 0.0001) [7]. Within the study case sample, the most prevalent Duffy genotypes were FYA/A (43.9%) and FYA/B (44.1%), while the FYB/B genotype was present in 11.9% of cases. It is important to note that these genotypes showed no differences between cases and controls and thus were not independently associated with malaria odds, regardless of parasite species stratification. Genotypes associated with reduced expression of the FYB allele were more frequently observed in malaria patients (16.7%; *n* = 152/909; *p* = 0.19), particularly in those with *P. vivax* malaria (17.7%; *n* = 112/633; *p* = 0.09), compared to healthy controls (14.5%; *n* = 132/909). When assessing the hospitalization rates among Duffy genotypes, it was determined that among the cases, 3.5% (*n* = 32/909) and 3.8% (*n* = 35/909) of individuals were hospitalised with severe malaria, respectively. The proportion of hospitalised patients was higher in FYA/A individuals (5.0%; *n* = 20/399), lower in FYA/B (3.5%; *n* = 14/401, *p* = 0.29), and significantly lower in FYB/B individuals (0.9%; *n* = 1/109, *p* = 0.06). Duffy blood group negativity was observed in 0.3% of cases [7].

Another study conducted in the borders in Thailand, Myanmar, and Malaysia collected samples from 1100 malaria cases and 1100 healthy subjects. Among them, 200 samples tested positive for *P. falciparum* and 900 tested positive for *P. vivax*. The FYA/A genotype was identified in 83.5% of patients and 75.0% of healthy individuals, while FYA/B was observed in 13% of patients with *P. vivax* and 24% of healthy individuals. FYB/B was detected in 3.5% of patients with *P. vivax* and 1% of healthy individuals. None of the study participants exhibited the FYAES/B^ES^ blood group, indicating the Fy(a−b−) phenotype. The frequency of FYA/A was significantly higher in patients with a *P. vivax* infection compared to healthy subjects (*p* = 0.036). In contrast, the frequency of FYA/B was significantly higher in healthy subjects compared to those infected with *P. vivax* (*p* = 0.005). Although FYB/B was more prevalent in patients with *P. vivax*, the difference was not statistically significant. The FY gene mutation at nucleotide position 265 could not be confirmed in the 167 samples that tested negative for FYB [32].

In Iran, the analysis results revealed that the most common Duffy genotype in the cases was FYA/B (*n* = 83; 51.9%), followed by FYA/A (*n* = 26; 16.3%), FYB/B (*n* = 22; 13.8%), FYA/B^ES^ (*n* = 16; 10%), FYB/B^ES^ (*n* = 11; 6.9%), and FYB^ES^/B^ES^ (*n* = 2; 1.3%). The predominant phenotype in the case group was Fy(a+b+) at 51.9%, similar to the controls but with a slightly lower percentage of 41.3% [25].

In Latin America, particularly in Haiti, the presence of the FYB^ES^ allele was demonstrated in 99.4% (*n* = 163/164) of *P. vivax* cases [24]. In Colombia, a study involving 320 volunteers was conducted, with women accounting for 59% of the participants. Among the volunteers, 73 individuals (23%) were Afro-Colombians, 74 (23%) were indigenous natives, and 173 (54%) were mestizos. Malaria was detected in 17% of the participants (52 out of 320). The T-46 allele frequency ranged from 90% to 100%, while among Afro-Colombians, it was 50%. At the 131 loci, the maximum frequency of the G allele was 30% in Amerindians, and the maximum frequency of the A allele was 69% in Afro-Colombians. The results revealed that there was an absence of Duffy-negative individuals infected with *P. vivax* [16]. In Brazil, there have been reports of four studies. The first study diagnosed *P. vivax* malaria in 225 patients, of whom 52.4% were men. Among them, 97 had uncomplicated malaria, while 128 had severe malaria. The distribution of the Duffy genotype/phenotype was as follows: 96 individuals had FYA/FYB, Fy(a+b+), 36 patients had FYA/FYB^ES^, Fy(a+b−), 36 individuals had FYB/FYB, Fy(a−b+), 34 had FYA/FYA, Fy(a+b−), 20 individuals had FYB/FYB^ES^, Fy(a−b+), 2 had FYA/FYAW, Fy(a+w), and 1 had FYB^ES^/FYB^ES^, Fy(a−b−) [44]. The second study evaluated 287 individuals (70% men) who were experiencing their initial diagnosis of *P. vivax* malaria without co-infection with other *Plasmodium* species or comorbidities with FYA/FYA, observed in 63 subjects (23.7%), FYA/FYB in 114 individuals (42.8%), FYA/FYB^ES^ in 23 individuals (8.6%), FYA/FYBW in 01 individual (0.4%), FYB/FYB in 55 individuals (20.7%), FYB/FYB^ES^ in 08 individuals (3%), FYB/FYBW in 01 individual (0.4%), and FYB^ES^/FYBW in 01 individual (0.4%) [28]. The third study analysed blood samples from 690 individuals with a median age of 25. The aim was to assess the potential influence of DARC on susceptibility to clinical *P. vivax* malaria. The number of malaria episodes over seven years (2003–2009) was recorded. Although the median number of episodes was 0, a significant variation ranging from 0 to 24 was observed. The prevalence of malaria was 7% (*n* = 35/498), with 89% of the cases being attributed to *P. vivax.* The Duffy genotype FYA/FYB was present in 29.6% of the population, FYA/FYA in 23.2%, and FYA/FYB^ES^ in 20.1%. Conversely, the FYB^ES^/FYB^ES^ genotype was observed in only 3% of cases. Overall, the study population showed a predominance of the functional DARC alleles FYA (48%) and FYB (33.6%). An adjusted Poisson regression analysis, considering the place of residence and duration of residence in the endemic area, revealed a 19% risk reduction (95% CI: 2%–32%; *p* = 0.029) for clinical *P. vivax* malaria in individuals with the FYA/FYB^ES^ genotype and a 91% risk reduction (95% CI: 67%–97%; *p* = 0.0003) in those with the FYB^ES^/FYB^ES^ genotype compared to individuals with the FYA/FYB genotype. Conversely, individuals with the FYB/FYB^ES^ genotype had a higher risk (26%; 95% CI: 3%–53%; *p* = 0.023) of clinical malaria compared to individuals with the reference genotype, FYA/FYB. Furthermore, susceptibility to malaria decreased among DARC genotypes with a longer duration of residence in the endemic area. Each additional year of residence in the endemic area resulted in a 3% reduction (95% CI: 2.5%–3.4%; *p* < 0.0001) in the risk of *P. vivax* malaria [27]. The final study, conducted in the Marajó Archipelago situated east of the Amazon, involved the analysis of 678 individuals, and the presence of *Plasmodium* was detected in 137 samples, corresponding to 20.2% of the total samples analysed. The prevalence of *P. vivax* was determined to be 13.9% (*n* = 94/678), while *P. falciparum* accounted for 5.8% (*n* = 39/678) of the cases. Additionally, there were cases of co-infection involving *P. falciparum*/*P. vivax*, which represented 0.6% (*n* = 4/678) of the cases. The study revealed that 4.3% (*n* = 29/678) of the patients included in the study were categorised as Duffy-negative (FYB^ES^/FYB^ES^), whereas 95.7% (*n* = 649/678) were classified as Duffy-positive. Among the individuals with a Duffy-negative status, 6.9% (*n* = 2/29) presented *P. vivax* malaria, whereas the prevalence was 14.7% among those with a Duffy-positive status. The risk of *P. vivax* malaria occurrence in Duffy-negative individuals was lower, although not statistically significantly lower, compared to Duffy-positive individuals (OR 0.4460; 95% CI: 0.1044–1.9060; *p* = 0.3983) [22].

## 4. Discussion

Malaria remains the most important vector-borne parasitic disease in the world, with *P. vivax* being one the most important species according to its prevalence in some areas of the globe. The evolution of the changes in genotype and phenotype expression trends over time in various regions of continents with endemic areas, including Africa, Asia, and Latin America, should be emphasised. Notably, an association was observed between the Duffy-positive genotype and a higher incidence of malaria infection. Nevertheless, despite the heterogeneous nature of the results, a clear trend indicating the lack of protective effect of the Duffy-negative genotype against malaria was evident. However, thoroughly analysing all the variables is necessary for a more comprehensive conclusion.

Shifting the focus to the Duffy genotype/phenotype relationship and *P. vivax* invasion, studies have reported a predominance of the Duffy-negative genotype FY*B^ES^/*B^ES^. Yet, no cases of *P. vivax* malaria were documented [10,24]. In contrast to the reported cases in Ethiopia, Sudan, Cameroon, Madagascar, and Brazil, where Duffy-negative subjects had a range of *P. vivax* infection proportions from 0.8% to 6.9% [21,22,32,36,38,42], the risk of malaria for Duffy-negative hosts in Madagascar was half that of Duffy-positive individuals (OR 0.52) [36]. Similarly, in Brazil, Duffy-negative subjects had a lower but not significantly different risk of presenting malaria caused by *P. vivax* compared to Duffy-positive individuals (OR 0.44; *p* = 0.3983) [22]. Additional studies by Hoque et al. [42] and Keple et al. [11] demonstrated a higher prevalence of *P. vivax* malaria in Duffy-negative subjects, with rates of 16.7% and 14.95%, respectively. Three notable studies conducted in vulnerable areas with minimal resources drew attention due to their high prevalence (in Duffy-negative subjects), ranging from 86% to 99% [33,35,40].

Only one systematic review has investigated the prevalence of *P. vivax* malaria in Duffy-negative individuals [46,47,48,49,50,51,52,53]. This review revealed that 100% of Duffy-negative subjects had reported cases of *P. vivax* malaria in 11 studies. These studies were conducted in various regions of Africa, including West Africa (Nigeria, Senegal, Mali, and Benin), Central Africa (Cameroon, Angola, and Equatorial Guinea), North Africa (Sudan), and East Africa (Kenya). Additionally, a meta-analysis of 14 studies showed that the combined prevalence of *P. vivax* infection in Duffy-negative subjects was 25%. Furthermore, a meta-regression analysis was performed to determine the significance of the African continent as a covariate and a source of heterogeneity, which was found to be significant [53]. Notably, both this meta-analysis [53] and our systematic review found that the reported prevalences are higher than those described in high-risk populations, such as pregnant women, who have been observed to have figures up to 11.1% using a random-effects model [54], resulting in a combined prevalence of 4.5%. Similarly, the prevalence of congenital malaria is only 6.9% [55].

Finally, studies have reported a significantly higher frequency of the FYA/A genotype in *P. vivax*-infected patients than in healthy individuals. In comparison, healthy individuals showed a significantly higher frequency of FYA/B than those infected with *P. vivax* [28,32], indicating a highly heterogeneous distribution of Duffy genotypes and phenotypes. However, in Iran, malaria cases predominantly exhibited the FYA/B genotype (51.9%), followed by FYA/A (16.3%) [25]. In Brazil, the main genotypes/phenotypes identified among the cases were FYA/B, Fy(a+b+), FYA/B^ES^ Fy(a+b), and to a lesser extent, FYB^ES^/B^ES^ Fy(a−b−) [27,44]. When comparing these results with the evidence, a meta-analysis by Wilairatana et al. [53] revealed that the Duffy genotype negativity acted as a protective factor against *P. vivax* infection in studies conducted in Sudan, Madagascar, Ethiopia, and Mauritania. Only one study conducted in Brazil showed a higher risk of infection among Duffy-negative individuals. However, four studies conducted in Cameroon, Ethiopia, Sudan, and Iran identified no differences in infection risk. The combined analysis demonstrated decreased odds of *P. vivax* malaria among Duffy-negative individuals (OR 0.46; 95% CI: 0.26–0.82; *p* = 0.009) [53]. These findings correlate with differences in the evolution pattern of *Plasmodium* and humans, as observed in the global variation of genotypes and phenotypes. For instance, in Saudi Arabia, the FYA and FYB antigen frequencies were 12.58% and 11.18%, respectively. The Fy phenotypes were distributed as follows: Fy(a+b−), 15 (10.48%); Fy(a−b+), 13 (9.10%); Fy(a+b+), 3 (2.10%); and Fy(a−b−), 112 (78.32%) [56]. On the other hand, in Colombia, the Fy(a−b−) phenotype had the highest prevalence (48%), followed by the Fy(a−b+) phenotype, the Fy(a+b−) phenotype, and to a lesser extent, the Fy(a+b+) phenotype [57]. Continuous research is necessary to assess the risks and propose strategies from a translational perspective that enable the development of drugs or effective and sustainable interventions over time. This research should focus on studying the disease distribution and understanding the expression of the Duffy genotype and phenotype in humans.

Regarding limitations, we must acknowledge that we identified a limited number of studies that report *P. vivax* infection among Duffy-negative individuals. Additionally, there is insufficient scientific evidence concerning the relationship between the Duffy genotype/phenotype and infection by other Plasmodium species. Secondly, the studies exhibit high heterogeneity, making it challenging to organise the information and potentially causing confusion due to the topic’s density and complexity. Thirdly, many studies employ a descriptive observational methodology, which hinders the extrapolation of associations that could enhance our understanding of the relationship between the parasites’ Duffy-binding protein and erythroid cell Duffy antigens. Consequently, it is not easy to establish causal principles. Furthermore, the obtained information could not be meta-analysed due to the high heterogeneity in the reporting of genotypes and phenotypes in the included studies, as well as significant variations in the methodology utilised across these studies. Nonetheless, this systematic review represents one of the first comprehensive evaluations providing valuable and novel evidence on the relationship between Duffy genotype/phenotype and *P. vivax* prevalence.

## 5. Conclusions

This systematic review found that the homozygous Duffy genotype positive for the A allele (FY*A/*A) is associated with a higher incidence of *P. vivax* infection. Furthermore, the negative Duffy genotype does not confer protection against *P. vivax* malaria.

## Figures and Tables

**Figure 1 tropicalmed-08-00463-f001:**
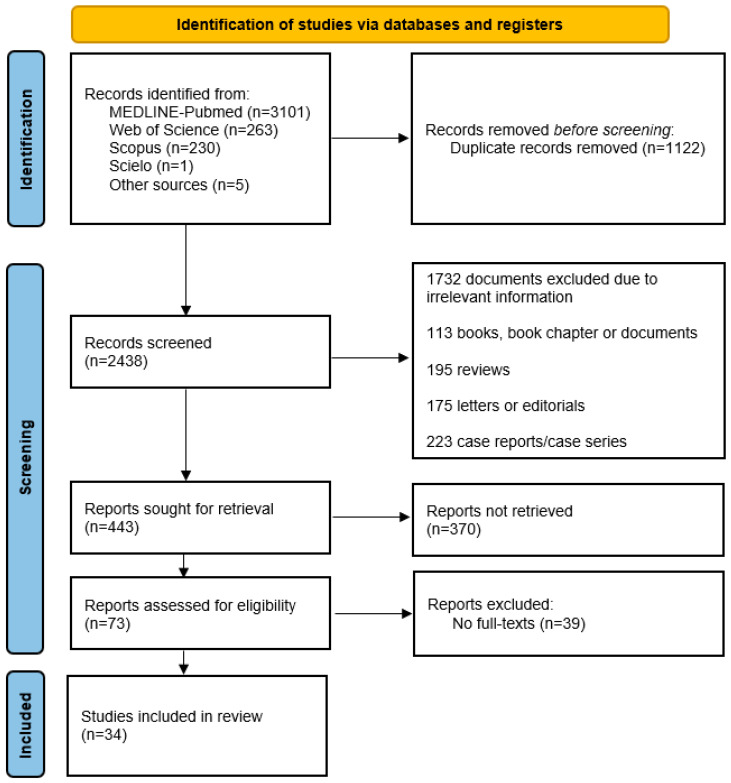
Characteristics of studies evaluating the Duffy-negative genotype/phenotype and prevalence of *Plasmodium vivax* infection. Initially, a total of 3600 documents were identified, which, subsequent to the elimination of duplicates and the application of eligibility criteria, comprised a final total of 34 published articles.

**Figure 2 tropicalmed-08-00463-f002:**
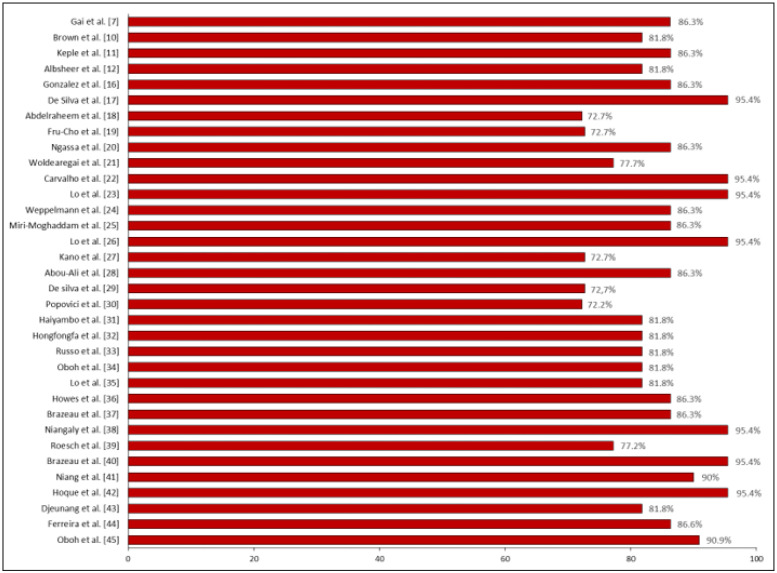
Percentage of compliance with reporting quality according to the STROBE tool [7,10,11,12,16,17,18,19,20,21,22,23,24,25,26,27,28,29,30,31,32,33,34,35,36,37,38,39,40,41,42,43,44,45].

**Table 1 tropicalmed-08-00463-t001:** General characteristics of studies that assessed the relationship between Duffy genotype/phenotype and *P. vivax* prevalence.

Authors	Country	Results	*P. vivax* Prevalence	Risk of Bias	Certainty	Significance
Brown et al. [10]	Ghana	-952 adults -Absence of FY*B^ES^ allele in 90.5% of the population -No cases of *P. vivax*	0% for negative Duffy	Serious	⨁⨁⨁◯ (Moderate)	Important
Oboh et al. [45]	Nigeria	-242 malaria cases -All were Duffy negative genotype	2.7% for negative Duffy	Serious	⨁⨁⨁◯ (Moderate)	Important
Ferreira et al. [44]	Brazil	-225 malaria cases -Fy(a+b−): 31.1% -Fy(a+b+): 42.7% -Fy(a−b+): 24.8% -Fy(a−b−): 0.44%	0.4% for Fy(a−b−)	Serious	⨁⨁⨁◯ (Moderate)	Important
Djeunang et al. [43]	Cameroon	-1001 malaria cases -181 caused by *P. vivax* with Duffynegative genotype	18% for negative Duffy	Serious	⨁⨁⨁◯ (Moderate)	Important
Hoque et al. [42]	Sudan	-42 malaria cases -83.3% Duffy-positive (10 homozygous/25 heterozygous)	16.7% for negative Duffy	Serious	⨁⨁⨁◯ (Moderate)	Important
Niang et al. [41]	Senegal	-74 malaria cases -Pure infection by *P. falciparum*: 79.7%	20.3% for negative Duffy	Non-serious	⨁⨁⨁⨁ (High)	Critical
Brazeau et al. [40]	Democratic Republic of Congo	-172 infections by *P. vivax* -14 infections in Duffy-negative individuals	8.3% for negative Duffy	Serious	⨁⨁⨁◯ (Moderate)	Important
Roesch et al. [39]	Cambodia and Madagascar	-174 malaria cases -T/T substitution in 100% in Cambodia/44% T/T—56% T/C in Madagascar	100% for positive Duffy	Serious	⨁⨁⨁◯ (Moderate)	Important
Niangaly et at. [38]	Mali	-Screening of 300 children -1 to 3 cases per 25 Duffy-negative children	-	Non-serious	⨁⨁⨁⨁ (High)	Critical
Albsheer et al. [12]	Sudan	-992 samples -190 infections by *P. vivax* (Fy(a−b+): 67.9%/Fy(a+b−): 14.2%/Fy(a−b−): 17.9%	67.9% Fy(a−b+)/17.9% Fy(a−b−)	Non-serious	⨁⨁⨁⨁ (High)	Critical
Brazeau et al. [37]	Democratic Republic of Congo	-17,972 samples -579 infections by *P. vivax* and 467 sequencings (*n* = 464/467 for Duffy-negative)	99.3% for negative Duffy	Serious	⨁⨁⨁◯ (Moderate)	Important
Howes et al. [36]	Madagascar	-1878 adults -48.7% Duffy-negative -86 and 44 infections by *P. vivax* with Duffy-positive and Duffy-negative, respectively	8.9% for negative Duffy/4.8% for positive Duffy	Serious	⨁⨁⨁◯ (Moderate)	Important
Kepple et al. [11]	Sudan and Ethiopia	-107 and 305 individuals infected with *P. vivax* for Duffy-negative and Duffy-positive	14.95% for negative Duffy/13.77% for positive Duffy	Serious	⨁⨁⨁◯ (Moderate)	Important
Lo et al. [35]	Sudan and Ethiopia	-1963 samples -332 infections by *P. vivax* (49 for Duffy-negative)	9.2%–86% for negative Duffy	Serious	⨁⨁⨁◯ (Moderate)	Important
Oboh et al. [34]	Nigeria	-436 samples and 256 cases -5 infections by *P. vivax* (all Duffy-negative homozygotes)	1.95% for negative Duffy	Serious	⨁⨁⨁◯ (Moderate)	Important
Russo et al. [33]	Cameroon	-484 samples -27 infections by *P. vivax* (all Duffy-negative)	5.6% for negative Duffy	Serious	⨁⨁⨁◯ (Moderate)	Important
Hongfongfa et al. [32]	Thailand and Myanmar	-900 cases of *P. vivax* -FY*A/*A: 83.5% of cases	0% for Fy(a−b−)	Non-serious	⨁⨁⨁⨁ (High)	Critical
Haiyambo et al. [31]	Namibia	-33 cases and 47 controls -3 infections by *P. vivax* (all Duffy-negative)	9% for negative Duffy	Non-serious	⨁⨁⨁⨁ (High)	Critical
Popovici et al. [30]	Cambodia	-22 Duffy-positive cases (16 FY*A/*A homozygotes)	-	Serious	⨁⨁⨁◯ (Moderate)	Important
Gai et al. [7]	India	-909 malaria cases (43.9% FY*A/A vs. 44.1% FYA/*B) -633 infections by *P. vivax* (44.2% FY*A/A vs. 43.7% FYA/*B)	0.3% for negative Duffy	Non-serious	⨁⨁⨁⨁ (High)	Critical
De Silva et al. [29]	Malaysia	-79 infections by *P. knowlesi* -Equal distribution of FY*A/A and FYA/*B genotypes	-	Non-serious	⨁⨁⨁⨁ (High)	Critical
Abou-Ali et al. [28]	Brazil	-287 infected by *P. vivax* -23.7% FYA/FYA; 42.8% FYA/FYB; 3% FYB/FYB	-	Serious	⨁⨁⨁◯ (Moderate)	Important
Kano et al. [27]	Brazil	-Reduction in risk of clinical *P. vivax* malaria by 19% and 91% for FYA/B^ES^ and FYB^ES^/B^ES^ genotypes, compared to FYA/*B	-	Serious	⨁⨁⨁◯ (Moderate)	Important
Lo et al. [26]	Ethiopia	-145 symptomatic individuals infected by *P. vivax* -69.7% FY*A/B^ES^ or FYB/*B^ES^ -1.4% FY*B^ES^/*B^ES^ (Duffy negative homozygotes)	-	Serious	⨁⨁⨁◯ (Moderate)	Important
Miri- Moghaddam et al. [25]	Iran	-160 infections by *Plasmodium* -FY*A/*B: 51.9% -FY*A/*A: 16.3% -FY*B/*B: 13.8% -FY*A/*B^ES^: 10%	0.6% for negative Duffy	Non-serious	⨁⨁⨁⨁ (High)	Critical
Weppelmann et al. [24]	Haiti	-164 cases -99.4% FY^ES^ allele	-	Serious	⨁⨁⨁◯ (Moderate)	Important
Lo et al. [23]	Ethiopia	-416 samples and 94 cases for Duffy-negative -3 cases of *P. vivax* in Duffy negative	3.1% for negative Duffy	Non-serious	⨁⨁⨁⨁ (High)	Critical
Carvalho et al. [22]	Brazil	-678 cases and 94 infections by *P. vivax* -29 Duffy-negative individuals (2 cases of *P. vivax*)	6.9% for negative Duffy	Serious	⨁⨁⨁◯ (Moderate)	Important
Woldearegai et al. [21]	Ethiopia	-1931 adults -111 cases of *P. vivax*	20% for negative Duffy	Serious	⨁⨁⨁◯ (Moderate)	Important
Ngassa et al. [20]	Cameroon	-201 symptomatic cases -8 cases of *P. vivax* infection	3.9% for negative Duffy	Serious	⨁⨁⨁◯ (Moderate)	Important
Fru-Cho et al. [19]	Cameroon	-87 malaria cases -12 infections by *P. vivax* (6 in Duffy-negative individuals)	6.8% for negative Duffy	Serious	⨁⨁⨁◯ (Moderate)	Important
Abdelraheem et al. [18]	Sudan	-126 suspected cases -48 confirmed cases of *P. vivax* (4 in Duffy-negative individuals)	8.3% for negative Duffy	Serious	⨁⨁⨁◯ (Moderate)	Important
De Silva et al. [17]	Malaysia	-111 samples -Fy(a+b−): 89.2% -FY*A/*A: 48 cases	0% for negative Duffy	Non-serious	⨁⨁⨁⨁ (High)	Critical
Gonzalez et al. [16]	Colombia	-52 individuals infected by *Plasmodium* (14 with *P. vivax*) -Amerindians and mestizos: T-46 allele in 90%–100%/Afro-Colombians 50%	-	Serious	⨁⨁⨁◯ (Moderate)	Important

## Data Availability

Not applicable.

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
