# Peer review of "Relationship between Duffy Genotype/Phenotype and Prevalence of Plasmodium vivax Infection: A Systematic Review"

_tropicalmed, 2023, doi:10.3390/tropicalmed8100463_

Round 1

Reviewer 1 Report

Thank you for your informative systematic review. Some authors believe that those individuals that genetically are duffy blood negative they do not get vivex malaria. But some others do not believe such phenomenon. Authors of the manuscript have tried to gather the different relevant studies as much as possible in a systematic review. The topic is relevant in the field. It reviewing most of the relevant published materials. Compared with other published material, the systematic review methodologies are almost the same. Although your review indicates that Duffy negative genotype does not confer protection against vivax malaria, more studies are needed to confirm your conclusion.

Author Response

Comments to Reviewer 1

Thank you for your informative systematic review. Some authors believe that those individuals that genetically are Duffy blood negative they do not get vivax malaria. But some others do not believe such phenomenon. Authors of the manuscript have tried to gather the different relevant studies as much as possible in a systematic review. The topic is relevant in the field. It reviewing most of the relevant published materials. Compared with other published material, the systematic review methodologies are almost the same. Although your review indicates that Duffy negative genotype does not confer protection against vivax malaria, more studies are needed to confirm your conclusion.

Answer: Thanks a lot for your positive comments.

Reviewer 2 Report

This is a review on the relationship between Duffy genotype/phenotype and P. vivax infection, in particular on the question whether Duffy-negative persons are naturally protected from P. vivax infection. After the selection process, 34 published articles were analyzed.

Introduction: The introductory section provides sufficient background information.

Methods: The authors say that they followed the standard methodological procedures for systematic reviews, including registration (PROSPERO), PRISMA, STROBE, and GRADE tools. Few details concerning article selection should be checked for further clarity. The screening procedure (Fig. 1) needs more explanations. Please add a legend to this figure.

Results: There is a lot of redundancy. The purpose of a review paper is not to provide a detailed description of each of 34 selected papers. I think that the authors should focus on the pertinent data in the selected papers and give a synthetic view of all 34 papers. Please see my “major comments.”

Results: There is a lot of redundancy.

Discussion: At least 3 paragraphs in the Discussion section are not pertinent to this review paper. Please see my “major comments.”

Please italicize all Latin names (example, P. vivax).

MAJOR COMMENTS:

Results and Organization of the paper: The authors describe their results, article by article, in sections 3.3 and 3.4, as well as in Table 1. The text in sections 3.3 and 3.4 summarizes in detail the results obtained in each of 34 articles. In fact, each of 34 articles is described three times: once in section 3.3, second time in section 3.4, and the third time in the table. In short, there is a lot of redundancy in this paper. The authors should considerably shorten the text in sections 3.3 and 3.4, combine them to avoid repeating the description of each study in each section, and provide the essential results of each study in the table.  

Discussion: The authors digress into irrelevant factors in lines 488-530 (sex, age, ethnic origin, bed net use). The text in these lines does not concern the topic of this review paper: Duffy vs P. vivax. These lines can be deleted.

Line 600-601, Conclusions: Unlike G6PD, Duffy is not an X-linked condition. There seems to be no reason why a review paper should look for a gender-associated correlation between Duffy and P. vivax infection.  

Line 603-604, Conclusions, “negative Duffy genotype does not confer protection against this disease”: The statement is not clear, especially what “this disease” is referring to: malaria in general or P. vivax malaria? If referring to P. falciparum, P. ovale, or P. malariae, the information is not new. If referring to P. vivax alone, the conclusion of this review paper is somewhat disappointing because we all know, since more than a decade, that Duffy-negative individuals can be infected with P. vivax.

MINOR COMMENTS:

Lines 53-54: CDC does not maintain the database on global malaria cases. The data cited by the authors come from the WHO. Please cite the appropriate sources (World Malaria Report 2022) for reference 2.

Lines 54-55, “United States”: “United States of America”

Lines 54-57: The authors are talking about imported malaria in the United States of America, but cite a reference that pertains to imported malaria in Europe (ref 3). Please cite an appropriate reference to support the statement.

Line 58, “Plasmodium”: in italics

Line 59, “leading to potentially fatal alterations”: This expression is not clear. Please rewrite it.

Lines 59-60, “The invasion occurs when infected female Anopheles mosquitoes bite humans”: Not “invasion” but “infection”

Lines 59-60, Anopheles: in italics

Line 61, “can cause pathology in humans”: This is not always true with some Plasmodium species. An infected person can remain an asymptomatic carrier for years without any obvious pathology.

Line 66, IL-8: interleukin-8 (IL-8)

Line 65: It would be helpful to remind some readers who may not be familiar with the classification of chemokines that they are one of the subgroups of cytokines and that they are classified based on the number and arrangement of cysteine: 4 cysteine group (CC motif, CXC motif, CX3C motif with 3 amino acids between cysteine residues; 2 cysteine group (XC chemokines, etc.

Line 70, “the interaction with DARC is essential for the invasion of Plasmodium vivax”: The interaction with DARC is thought to be essential for the invasion of Plasmodium vivax (in italics)…the absence of this protein has been thought to be a preventive factor…

Line 76, “infestation”: parasite infection

Line 77, “data…exists”: exist

Line 78, “to achieve the WHO’s goal”: The authors refer to WHO but cite CDC (ref 8). Please cite a WHO document here to be coherent.

Line 80 and elsewhere in the text, “P. vivax infestation”: infection

Line 81, “more than 1/3 of the global population is exposed to it [P. vivax]”: Ref 9 is not the most appropriate citation here. Please cite a paper on the epidemiology of P. vivax malaria here.

Line 83, “Duffy antigen is essential for entering this Plasmodium species…”: is essential for invasion of P. vivax parasites into reticulocytes

Lines 81-84, “cases of P. vivax in Duffy-negative”: Ref 10 does not seem to support this statement. Please check.

Lines 101-103 versus line 114: The authors state in lines 101-103 that “filters were applied to select case reports,…” and state in line 114 that “full-text articles were considered.” These two statements sound contradictory. Please clarify.

Figure 1: At the first step of the screening procedure, it is not clear how the authors got 2438 (screened) – 2035 (excluded) = 443 (sought for retrieval). 2438-2035 = 403. A figure caption/legend should be added to explain what the double asterisks (**) refer to. The reasons for “Reports not retrieved (n = 370)” should be given. For “Reports excluded: no full-texts (n = 39)” further explanations should be provided. Is it because access to the articles is not free? Any other reasons why they were excluded?

Table 1: The authors refer to Table 1 in line 145. Table 1 is placed in line 289. It should be placed before Figure 2 for easier reference for the readers.

Line 181, “mixed P. vivax/P infections. falciparum in 33 samples”: P. vivax/P. falciparum mixed infections

Line 244, “In Namibia..”: I would start a new paragraph here.

Line 265, “another study conducted in the borders between Thailand, Myanmar, and Malaysia”: It is not clear in which country this study was conducted. Please specify the country. Also, I would start a new paragraph here to separate it from the preceding paragraph on India.

Lines 150-152, “Ghana”: In section 3.3, the authors describe, country by country, continent by continent, the essential findings of 34 studies included in this review paper. Why is the study performed in Ghana described so succinctly, in only 2 lines, whereas most other studies received more attention, usually with several lines or even one whole paragraph? I think that the text needs to be balanced. In any case, please refer to my “major comments” concerning redundancy.

Line 370, Dschang: There is a need for a transition. Lines 367-370 concern Nigeria. From line 370, the authors are talking about Cameroon.

Lines 539-541, “Additional studies by Hoque et al. and Keple et al. demonstrated a higher prevalence of P. vivax malaria in Duffy-negative subjects, with rates of 16.7% and 14%, respectively”: Please complete this sentence. What were the prevalence rates of P. vivax in Duffy-positive subjects in these two studies?

Line 574, “Arabia”: Saudi Arabia or Arabian peninsula?

Lines 602-603, “associated with a higher incidence of Plasmodium infection”: Plasmodium species in general or P. vivax?

Author Response

Comments to Reviewer 2

This is a review on the relationship between Duffy genotype/phenotype and P. vivax infection, in particular on the question whether Duffy-negative persons are naturally protected from P. vivax infection. After the selection process, 34 published articles were analyzed.

-Introduction: The introductory section provides sufficient background information.

Answer: We thank the peer reviewer for their comment with the aim of improving the quality of our manuscript. We have improved our introduction.

-Methods: The authors say that they followed the standard methodological procedures for systematic reviews, including registration (PROSPERO), PRISMA, STROBE, and GRADE tools. Few details concerning article selection should be checked for further clarity. The screening procedure (Fig. 1) needs more explanations. Please add a legend to this figure.

Answer: We thank the peer reviewer for their comment with the aim of improving the quality of our manuscript. Corrected.

-Results: There is a lot of redundancy. The purpose of a review paper is not to provide a detailed description of each of 34 selected papers. I think that the authors should focus on the pertinent data in the selected papers and give a synthetic view of all 34 papers. Please see my “major comments.”

Answer: We thank the peer reviewer for their comment with the aim of improving the quality of our manuscript. Corrected.

-Results: There is a lot of redundancy.

Answer: We thank the peer reviewer for their comment with the aim of improving the quality of our manuscript. Corrected.

-Discussion: At least 3 paragraphs in the Discussion section are not pertinent to this review paper. Please see my “major comments.”

Answer: We thank the peer reviewer for their comment with the aim of improving the quality of our manuscript. Corrected.

-Please italicize all Latin names (example, P. vivax).

Answer: We thank the peer reviewer for their comment with the aim of improving the quality of our manuscript. Corrected.

 MAJOR COMMENTS:

- Results and Organization of the paper: The authors describe their results, article by article, in sections 3.3 and 3.4, as well as in Table 1. The text in sections 3.3 and 3.4 summarizes in detail the results obtained in each of 34 articles. In fact, each of 34 articles is described three times: once in section 3.3, second time in section 3.4, and the third time in the table. In short, there is a lot of redundancy in this paper. The authors should considerably shorten the text in sections 3.3 and 3.4, combine them to avoid repeating the description of each study in each section, and provide the essential results of each study in the table. 

Answer: We thank the peer reviewer for their comment with the aim of improving the quality of our manuscript. Corrected.

- Discussion: The authors digress into irrelevant factors in lines 488-530 (sex, age, ethnic origin, bed net use). The text in these lines does not concern the topic of this review paper: Duffy vs P. vivax. These lines can be deleted.

Answer: We thank the peer reviewer for their comment with the aim of improving the quality of our manuscript. Corrected.

- Line 600-601, Conclusions: Unlike G6PD, Duffy is not an X-linked condition. There seems to be no reason why a review paper should look for a gender-associated correlation between Duffy and P. vivax infection. 

Answer: We thank the peer reviewer for their comment with the aim of improving the quality of our manuscript. Corrected.

- Line 603-604, Conclusions, “negative Duffy genotype does not confer protection against this disease”: The statement is not clear, especially what “this disease” is referring to: malaria in general or P. vivax malaria? If referring to P. falciparum, P. ovale, or P. malariae, the information is not new. If referring to P. vivax alone, the conclusion of this review paper is somewhat disappointing because we all know, since more than a decade, that Duffy-negative individuals can be infected with P. vivax.

Answer: We thank the peer reviewer for their comment with the aim of improving the quality of our manuscript. Corrected.

 MINOR COMMENTS:

 - Lines 53-54: CDC does not maintain the database on global malaria cases. The data cited by the authors come from the WHO. Please cite the appropriate sources (World Malaria Report 2022) for reference 2.

Answer: We thank the peer reviewer for their comment with the aim of improving the quality of our manuscript. Corrected.

- Lines 54-55, “United States”: “United States of America”

Answer: We thank the peer reviewer for their comment with the aim of improving the quality of our manuscript. Corrected.

- Lines 54-57: The authors are talking about imported malaria in the United States of America, but cite a reference that pertains to imported malaria in Europe (ref 3). Please cite an appropriate reference to support the statement.

Answer: We thank the peer reviewer for their comment with the aim of improving the quality of our manuscript. Corrected.

- Line 58, “Plasmodium”: in italics

Answer: We thank the peer reviewer for their comment with the aim of improving the quality of our manuscript. Corrected.

- Line 59, “leading to potentially fatal alterations”: This expression is not clear. Please rewrite it.

Answer: We thank the peer reviewer for their comment with the aim of improving the quality of our manuscript. Corrected.

- Lines 59-60, “The invasion occurs when infected female Anopheles mosquitoes bite humans”: Not “invasion” but “infection”

Answer: We thank the peer reviewer for their comment with the aim of improving the quality of our manuscript. Corrected.

- Lines 59-60, Anopheles: in italics

Answer: We thank the peer reviewer for their comment with the aim of improving the quality of our manuscript. Corrected.

- Line 61, “can cause pathology in humans”: This is not always true with some Plasmodium species. An infected person can remain an asymptomatic carrier for years without any obvious pathology.

Answer: We thank the peer reviewer for their comment with the aim of improving the quality of our manuscript. Corrected.

- Line 66, IL-8: interleukin-8 (IL-8)

Answer: We thank the peer reviewer for their comment with the aim of improving the quality of our manuscript. Corrected.

- Line 65: It would be helpful to remind some readers who may not be familiar with the classification of chemokines that they are one of the subgroups of cytokines and that they are classified based on the number and arrangement of cysteine: 4 cysteine group (CC motif, CXC motif, CX3C motif with 3 amino acids between cysteine residues; 2 cysteine group (XC chemokines, etc.)

Answer: We thank the peer reviewer for their comment with the aim of improving the quality of our manuscript. Corrected.

- Line 70, “the interaction with DARC is essential for the invasion of Plasmodium vivax”: The interaction with DARC is thought to be essential for the invasion of Plasmodium vivax (in italics)…the absence of this protein has been thought to be a preventive factor…

Answer: We thank the peer reviewer for their comment with the aim of improving the quality of our manuscript. Corrected.

- Line 76, “infestation”: parasite infection

Answer: We thank the peer reviewer for their comment with the aim of improving the quality of our manuscript. Corrected in all cases

- Line 77, “data…exists”: exist

Answer: We thank the peer reviewer for their comment with the aim of improving the quality of our manuscript. Corrected.

- Line 78, “to achieve the WHO’s goal”: The authors refer to WHO but cite CDC (ref 8). Please cite a WHO document here to be coherent.

Answer: We thank the peer reviewer for their comment with the aim of improving the quality of our manuscript. Corrected.

- Line 80 and elsewhere in the text, “P. vivax infestation”: infection

Answer: We thank the peer reviewer for their comment with the aim of improving the quality of our manuscript. Corrected in all cases

 - Line 81, “more than 1/3 of the global population is exposed to it [P. vivax]”: Ref 9 is not the most appropriate citation here. Please cite a paper on the epidemiology of P. vivax malaria here.

Answer: We thank the peer reviewer for their comment with the aim of improving the quality of our manuscript. Corrected.

- Line 83, “Duffy antigen is essential for entering this Plasmodium species…”: is essential for invasion of P. vivax parasites into reticulocytes

Answer: We thank the peer reviewer for their comment with the aim of improving the quality of our manuscript. Corrected.

- Lines 81-84, “cases of P. vivax in Duffy-negative”: Ref 10 does not seem to support this statement. Please check.

Answer: We thank the peer reviewer for their comment with the aim of improving the quality of our manuscript. Corrected.

- Lines 101-103 versus line 114: The authors state in lines 101-103 that “filters were applied to select case reports,…” and state in line 114 that “full-text articles were considered.” These two statements sound contradictory. Please clarify.

Answer: We thank the peer reviewer for their comment with the aim of improving the quality of our manuscript. Corrected.

- Figure 1: At the first step of the screening procedure, it is not clear how the authors got 2438 (screened) – 2035 (excluded) = 443 (sought for retrieval). 2438-2035 = 403. A figure caption/legend should be added to explain what the double asterisks (**) refer to. The reasons for “Reports not retrieved (n = 370)” should be given. For “Reports excluded: no full-texts (n = 39)” further explanations should be provided. Is it because access to the articles is not free? Any other reasons why they were excluded?

Answer: We thank the peer reviewer for their comment with the aim of improving the quality of our manuscript. Corrected.

- Table 1: The authors refer to Table 1 in line 145. Table 1 is placed in line 289. It should be placed before Figure 2 for easier reference for the readers.

Answer: We thank the peer reviewer for their comment with the aim of improving the quality of our manuscript. Corrected.

- Line 181, “mixed P. vivax/P infections. falciparum in 33 samples”: P. vivax/P. falciparum mixed infections

Answer: We thank the peer reviewer for their comment with the aim of improving the quality of our manuscript. Corrected.

- Line 244, “In Namibia..”: I would start a new paragraph here.

Answer: We thank the peer reviewer for their comment with the aim of improving the quality of our manuscript. Corrected.

- Line 265, “another study conducted in the borders between Thailand, Myanmar, and Malaysia”: It is not clear in which country this study was conducted. Please specify the country. Also, I would start a new paragraph here to separate it from the preceding paragraph on India.

Answer: We thank the peer reviewer for their comment with the aim of improving the quality of our manuscript. Corrected.

- Lines 150-152, “Ghana”: In section 3.3, the authors describe, country by country, continent by continent, the essential findings of 34 studies included in this review paper. Why is the study performed in Ghana described so succinctly, in only 2 lines, whereas most other studies received more attention, usually with several lines or even one whole paragraph? I think that the text needs to be balanced. In any case, please refer to my “major comments” concerning redundancy.

Answer: We thank the peer reviewer for their comment with the aim of improving the quality of our manuscript. Corrected.

- Line 370, Dschang: There is a need for a transition. Lines 367-370 concern Nigeria. From line 370, the authors are talking about Cameroon.

Answer: We thank the peer reviewer for their comment with the aim of improving the quality of our manuscript. Corrected.

- Lines 539-541, “Additional studies by Hoque et al. and Keple et al. demonstrated a higher prevalence of P. vivax malaria in Duffy-negative subjects, with rates of 16.7% and 14%, respectively”: Please complete this sentence. What were the prevalence rates of P. vivax in Duffy-positive subjects in these two studies?

Answer: We thank the peer reviewer for their comment with the aim of improving the quality of our manuscript. Corrected.

- Line 574, “Arabia”: Saudi Arabia or Arabian peninsula?

Answer: We thank the peer reviewer for their comment with the aim of improving the quality of our manuscript. Corrected.

- Lines 602-603, “associated with a higher incidence of Plasmodium infection”: Plasmodium species in general or P. vivax?

Answer: We thank the peer reviewer for their comment with the aim of improving the quality of our manuscript. Corrected.

Round 2

Reviewer 2 Report

The authors have considerably improved their manuscript. There are still a number of minor corrections that should be made.

Line 30, “Duffy negative phenotype as a protective factor against phenotypic malaria expression”: as a protective factor against Plasmodium vivax infection

Lines 34, 39, 42, and 44, “Plasmodium vivax”: P. vivax

Line 52, “The World malaria report 2022 indicate”: The WHO World Malaria Report 2022 suggested that there were 247...

Line 54, “predominantly due to imported episodes by travelers and immigrants...”: predominantly due to imported malaria

Lines 57-58, “when parasites from the Plasmodium group invade red blood cells, causing changes that could be life-threatening”: when parasites belonging to Plasmodium species invade and multiply in red blood cells. 

Line 61, “these species can also persist as asymptomatic carriers”: infected individuals can remain infected for years as asymptomatic carriers

Line 73, “into red blood cells/reticulocytes”: into reticulocytes [delete “red blood cells”]

Line 73, “African populations exhibit lower expression of the Duffy antigen”: Most African populations in sub-Saharan Africa either exhibit low expression of the Duffy antigen or are Duffy negative

Lines 94-96: Is the Duffy-negative genotype/phenotype a protective factor in the population susceptible to P. vivax infection compared to the Duffy-positive population?

Line 162, “Plasmodium Vivax”: Plasmodium vivax

Line 164, “34 documents”: 34 published articles

Lines 183-188, “FY*BES/BES”: “ES” should be in superscript for more clarity - FYBES/FYBES, FYA/ FYBES and FYB/ FYBES;  FYBES allele. Also FYBWK. Same remark elsewhere in the text (for example, lines 238, 239, 252, 254, 259, 341, 385, 394, 397, 409, 411, 443 and in the Discussion).

Line 214, “35”: Thirty-five

Line 264, “exhibited a negative phenotype”: exhibited a negative Duffy phenotype

Line 267, “9 14 cases”: Please check. There were only 292 samples. The total number of participants in DHS should be given.

Line 270, “rapid PCR”: What is a “rapid PCR”?

Lines 290-295: The diagnostic method for malaria detection should be cited (microscopy, rapid diagnostic test, or PCR?)

Lines 338-341, “Plasmodium invasions...(n=285).reported that...followed by FYA and FYB”: There is a problem with these sentences. Please correct them.

Lines 351-357, “In Asia...”: Please start a new paragraph here to separate the information reported from Cambodia from that of Madagascar.

Line 455, “Plasmodium general infection”: malaria infection

Lines 468-469, “a higher prevalence of P. vivax malaria in Duffy-negative subjects, with rates of 16.7% (vs 83.3% in Duffy-positive subjects)”: This is contradictory. 83.3% is higher than 16.7%.

Line 471, “due to their high prevalence, ranging from 86% to 99%”: This is not clear. Prevalence of what? (P. vivax? Duffy-negative? Duffy-positive?)

Line 516, “ other types of Plasmodia”: other Plasmodium species

Line 530, “Plasmodium vivax”: P. vivax

Author Response

Comments to Reviewer 2

-The authors have considerably improved their manuscript. There are still a number of minor corrections that should be made.

Answer: We thank the peer reviewer for their comment with the aim of improving the quality of our manuscript. All corrections were made.

-Line 30, “Duffy negative phenotype as a protective factor against phenotypic malaria expression”: as a protective factor against Plasmodium vivax infection

Answer: We thank the peer reviewer for their comment with the aim of improving the quality of our manuscript. Corrected

-Lines 34, 39, 42, and 44, “Plasmodium vivax”: P. vivax

Answer: We thank the peer reviewer for their comment with the aim of improving the quality of our manuscript. Corrected

-Line 52, “The World malaria report 2022 indicate”: The WHO World Malaria Report 2022 suggested that there were 247...

Answer: We thank the peer reviewer for their comment with the aim of improving the quality of our manuscript. Corrected

-Line 54, “predominantly due to imported episodes by travelers and immigrants...”: predominantly due to imported malaria

Answer: We thank the peer reviewer for their comment with the aim of improving the quality of our manuscript. Corrected

-Lines 57-58, “when parasites from the Plasmodium group invade red blood cells, causing changes that could be life-threatening”: when parasites belonging to Plasmodium species invade and multiply in red blood cells.

Answer: We thank the peer reviewer for their comment with the aim of improving the quality of our manuscript. Corrected

-Line 61, “these species can also persist as asymptomatic carriers”: infected individuals can remain infected for years as asymptomatic carriers

Answer: We thank the peer reviewer for their comment with the aim of improving the quality of our manuscript. Corrected

-Line 73, “into red blood cells/reticulocytes”: into reticulocytes [delete “red blood cells”]

Answer: We thank the peer reviewer for their comment with the aim of improving the quality of our manuscript. Corrected

-Line 73, “African populations exhibit lower expression of the Duffy antigen”: Most African populations in sub-Saharan Africa either exhibit low expression of the Duffy antigen or are Duffy negative

Answer: We thank the peer reviewer for their comment with the aim of improving the quality of our manuscript. Corrected

-Lines 94-96: Is the Duffy-negative genotype/phenotype a protective factor in the population susceptible to P. vivax infection compared to the Duffy-positive population?

Answer: We thank the peer reviewer for their comment with the aim of improving the quality of our manuscript. Corrected

-Line 162, “Plasmodium Vivax”: Plasmodium vivax

Answer: We thank the peer reviewer for their comment with the aim of improving the quality of our manuscript. Corrected

-Line 164, “34 documents”: 34 published articles

Answer: We thank the peer reviewer for their comment with the aim of improving the quality of our manuscript. Corrected

-Lines 183-188, “FY*BES/BES”: “ES” should be in superscript for more clarity - FYBES/FYBES, FYA/ FYBES and FYB/ FYBES;  FYBES allele. Also FYBWK. Same remark elsewhere in the text (for example, lines 238, 239, 252, 254, 259, 341, 385, 394, 397, 409, 411, 443 and in the Discussion).

Answer: We thank the peer reviewer for their comment with the aim of improving the quality of our manuscript. Corrected

-Line 214, “35”: Thirty-five

Answer: We thank the peer reviewer for their comment with the aim of improving the quality of our manuscript. Corrected

-Line 264, “exhibited a negative phenotype”: exhibited a negative Duffy phenotype

Answer: We thank the peer reviewer for their comment with the aim of improving the quality of our manuscript. Corrected

-Line 267, “9 14 cases”: Please check. There were only 292 samples. The total number of participants in DHS should be given.

Answer: We thank the peer reviewer for their comment with the aim of improving the quality of our manuscript. Corrected

-Line 270, “rapid PCR”: What is a “rapid PCR”?

Answer: We thank the peer reviewer for their comment with the aim of improving the quality of our manuscript. Corrected

-Lines 290-295: The diagnostic method for malaria detection should be cited (microscopy, rapid diagnostic test, or PCR?)

Answer: We thank the peer reviewer for their comment with the aim of improving the quality of our manuscript. Corrected

-Lines 338-341, “Plasmodium invasions...(n=285).reported that...followed by FYA and FYB”: There is a problem with these sentences. Please correct them.

Answer: We thank the peer reviewer for their comment with the aim of improving the quality of our manuscript. Corrected

-Lines 351-357, “In Asia...”: Please start a new paragraph here to separate the information reported from Cambodia from that of Madagascar.

Answer: We thank the peer reviewer for their comment with the aim of improving the quality of our manuscript. Corrected

-Line 455, “Plasmodium general infection”: malaria infection

Answer: We thank the peer reviewer for their comment with the aim of improving the quality of our manuscript. Corrected

-Lines 468-469, “a higher prevalence of P. vivax malaria in Duffy-negative subjects, with rates of 16.7% (vs 83.3% in Duffy-positive subjects)”: This is contradictory. 83.3% is higher than 16.7%.

Answer: We thank the peer reviewer for their comment with the aim of improving the quality of our manuscript. Corrected

-Line 471, “due to their high prevalence, ranging from 86% to 99%”: This is not clear. Prevalence of what? (P. vivax? Duffy-negative? Duffy-positive?)

Answer: We thank the peer reviewer for their comment with the aim of improving the quality of our manuscript. Corrected

-Line 516, “ other types of Plasmodia”: other Plasmodium species

Answer: We thank the peer reviewer for their comment with the aim of improving the quality of our manuscript. Corrected

-Line 530, “Plasmodium vivax”: P. vivax

Answer: We thank the peer reviewer for their comment with the aim of improving the quality of our manuscript. Corrected